# *Trichoderma* Biological Control to Protect Sensitive Maize Hybrids against Late Wilt Disease in the Field

**DOI:** 10.3390/jof7040315

**Published:** 2021-04-18

**Authors:** Ofir Degani, Shlomit Dor

**Affiliations:** 1Plant Sciences Department, Migal—Galilee Research Institute, Tarshish 2, Kiryat Shmona 11016, Israel; dorshlomit@gmail.com; 2Faculty of Sciences, Tel-Hai College, Upper Galilee, Tel-Hai 12210, Israel

**Keywords:** *Cephalosporium maydis*, corn, crop protection, fungus, *Harpophora maydis*, *Magnaporthiopsis maydis*, pathogenicity assay, real-time PCR

## Abstract

Late wilt, a disease severely affecting maize fields throughout Israel, is characterized by the relatively rapid wilting of maize plants from the tasseling stage to maturity. The disease is caused by the fungus *Magnaporthiopsis maydis*, a soil and seed-borne pathogen. The pathogen is controlled traditionally through the use of maize cultivars having reduced sensitivity to the disease. Nevertheless, such cultivars may lose their immunity after several years of intensive growth due to the presence of high virulent isolates of *M. maydis*. Alternative effective and economical chemical treatment to the disease was recently established but is dependent on the use of a dripline assigned for two adjacent rows and exposes the risk of fungicide resistance. In the current work, eight marine and soil isolates of *Trichoderma* spp., known for high mycoparasitic potential, were tested as biocontrol agents against *M. maydis*. An in vitro confront plate assay revealed strong antagonistic activity against the pathogen of two *T. longibrachiatum* isolates and of *T. asperelloides*. These species produce soluble metabolites that can inhibit or kill the maize pathogen in submerged and solid media culture growth assays. In greenhouse experiments accompanied by real-time PCR tracking of the pathogen, the *Trichoderma* species or their metabolites managed to improve the seedlings’ wet biomass and reduced the pathogen DNA in the maize roots. A follow-up experiment carried out through a whole growth session, under field conditions, provided important support to the *Trichoderma* species’ beneficial impact. The direct addition of *T. longibrachiatum* and even more *T. asperelloides* to the seeds, with the sowing, resulted in a yield improvement, a significant increase in the growth parameters and crops, to the degree of noninfected plants. These bioprotective treatments also restricted the pathogen DNA in the host tissues (up to 98%) and prevented the disease symptoms. The results encourage more in-depth research to uncover such biological agents’ potential and the methods to implement them in commercial fields. If adequately developed into final products and combined with other control methods, the biological control could play an important role in maize crop protection against Late wilt.

## 1. Introduction

Corn (*Zea mays*, maize) is considered one of the most important grain crops globally, with the United States and China being the top maize-producing countries [1]. In Israel, corn is a major crop in open areas, with a 3350 ha area harvested in 2019 and a volume production of 77,801 tons (FAOSTAT, 2019 Food and Agriculture Commodity Production data). Israel’s development of corn is under threat from severe disease—late wilt (LWD)—caused by the fungal pathogen *Magnaporthiopsis maydis* (synonyms *Cephalosporium maydis* (Samra, Sabet, and Hingorani) and *Harpophora maydis* [2,3]). LWD is characterized by the rapid wilt of sweet and fodder maize, mainly from the tasseling stage until shortly before maturity [4]. The pathogen is a hemibiotrophic, soil-borne [5], and seed-borne [6] fungus transmitted as spores, sclerotia, or hyphae on plant remains. *M. maydis* can survive for long periods in the soil or inside an alternative host plant, such as *Lupinus termis* L. (lupine) [7], *Gossypium hirsutum* L. (cotton), *Citrullus lanatus* (watermelon), and *Setaria viridis* (green foxtail) [8,9].

The pathogen disease mode in susceptible maize genotypes is well-detailed in the literature. According to Sabet et al. [10], the infection occurs during the first three weeks of the cultivation. After penetrating the roots, the pathogen first appears in the xylem three days after seeding (DAS) and spreads upwards. When the tassels first emerged (ca. 60 DAS), the fungus appeared throughout the stalk, the pathogen DNA levels peaked in the stems [11], and the first aboveground symptoms appeared shortly after that. About 10 days before harvest, the fungus hyphae and secreted materials blocked the water supply and led to the host’s rapid dehydration and death. The symptoms are enhanced under drought conditions [12,13]. In maize hybrids sensitive to LWD planted on heavily infested fields, LWD may cause 100% infection and total yield loss [4]. A similar infection mode (with some delay) occurs in resistant cultivars, and the pathogen can infect the seeds of these non-symptomatic plants and spread [11].

Today, there is a continuous need to develop novel strategies to study the pathogen, monitor its spread, and contain its devastating impact. The LWD is considered the most harmful disease in commercial maize fields in Egypt [14] and Israel [15]; poses a major threat in India [16], Portugal, and Spain [17]; and is a serious concern to other countries [18,19]. Attempts were made previously to control LWD using agricultural (balanced soil fertility and flood fallowing) [20,21], biological [22], physical (solar heating) [23], allelochemical [24], and chemical options [15,25,26], with different degrees of success. Lately, the tillage system’s effect and the cover crop have been demonstrated as important to the soil’s bioprotective role against *M. maydis* [27]. In 2018, for the first time since the discovery of LWD in Israel, an economical and efficient applicable solution was approved [12,28]. It can now be used on a large scale to protect the susceptible maize varieties.

Despite this recent encouraging success in developing control strategies against LWD, the rapid development of resistance to Azoxystrobin, the most effective antifungal compound against the late-wilt pathogen [4,12], may become a problem. Moreover, the extensive use of fungicides poses a serious concern; their residual effects and toxicity may affect the environment and human health [29]. Today, the most environmentally friendly, efficient, and cost-effective way to restrict the disease is by using highly resistant maize genotypes [16,30]. The recent years’ discovery of highly aggressive isolates of *M. maydis* [31,32] that may threaten resistant maize cultivars is forcing researchers to continue seeking alternative methods.

The LWD has been prevalent for about 40 years in North Israel—the Upper Galilee and especially in Hula Valley. In the last two decades, the disease has worsened and spread to the south of the country. It is now reported in most corn-growing areas in Israel. The extent of corn crops in metric ton yield per metric hectare in this country is exhibiting a constant upward trend, from 17.5 in 1987–1996 to 18.0 in 1997–2006 and 20.1 in 2007–2016 [33]. The effective risk management of LWD, primarily by avoiding sensitive maize cultivars’ growth, may contribute to this positive trend.

LWD biological control using *Trichoderma* spp. or other microorganisms has already been demonstrated in several studies (most recently, [22,34]). One of the most studied methods is using plant growth-promoting rhizobacteria as a biological control against LWD associated with the maize rhizosphere, which could also improve the plant health (summarized by [34]). Another approach is the use of species of the fungal genus *Trichoderma* against LWD. Species in this genus can form mutualistic endophytic relationships with several plant species [35]. Other species have been developed as biocontrol agents against fungal phytopathogens [36]. However, little information exists in the literature on the efficiency of *Trichoderma* spp. against the LWD. It was recently shown [22] that microalgae, *Chlorella vulgaris* extracts, with each of the *Trichoderma* species, *T. virens*, and *T. koningii*, were effective treatments against LWD under greenhouse and field conditions. These practices resulted in a 72% reduction in disease incidence in the greenhouse and a 2.5-fold higher grain production under field conditions. Still, the method should be inspected against the pathogen’s Israeli variants, and the action mechanism should be explored. Continuing to develop green solutions to efficiently control this pathogen for commercial grain production and maize seed production is an urgent need. 

The current research work aimed at examining the antimicrobial properties of *Trichoderma* species against *M. maydis*. The experiments set included various tests to study the selected *Trichoderma* species’ ability to decrease *M. maydis* growth. These practices included the confrontation assay, the *Trichoderma* spp. secreted metabolites’ effect on the pathogen’s development in enriched (solid and liquid) media, potted sprouts biocontrol assays, and full-growth season infection experiments under field conditions. A real-time PCR (qPCR)-based molecular detection of the pathogen DNA inside the host tissues was applied to test the treatments’ effectiveness.

## 2. Materials and Methods

### 2.1. Origin and Growth of the Fungus Magnaporthiopsis Maydis

One representative *M. maydis* isolate, designated Hm-2 (CBS 133165, deposited in the CBS-KNAW Fungal Biodiversity Center, Utrecht, The Netherlands), was selected for this study from our isolates’ library. This isolate was previously recovered from a cornfield of Kibbutz Sde Nehemia in Hula Valley in Upper Galilee, Northern Israel, in 2001 from Jubilee cv. corn plants showed dehydration symptoms. The Israeli *M. maydis* isolates were examined, characterized, and identified as described earlier [11]. The isolate was grown on potato dextrose agar (PDA; Difco Laboratories, Detroit, MI, USA) at a temperature of 28 ± 1 °C in the dark. These growth conditions allowed a high-humidity atmosphere inside the Petri dishes. Sowing the fungus into a new plate was done by transferring a 6-mm (in diameter) colony agar disk to a new PDA Petri dish. Fungus-containing agar disks were cut from the margins of a culture of *M. maydis*, which was grown on PDA for 4–6 days. Plates were labeled and incubated in a 28 ± 1 °C incubator in the dark. Growth in a liquid substrate was done using ten fungal disks sown in an Erlenmeyer flask containing 150-mL potato dextrose broth (PDB; Difco Laboratories, Detroit, MI, USA). The flasks were plugged with a breathable stopper and incubated for 6 days with shaking at 150 rpm at a temperature of 28 ± 1 °C in the dark.

### 2.2. Plate Confrontation Assay

A collection of *Trichoderma* spp. isolates (Table 1), most obtained from a marine source, were received courtesy of Prof. Oded Yarden (Hebrew University, Israel) and previously characterized [37]. Such marine *Trichoderma* isolates may be used in fields irrigated with water where the salinity percentage is relatively high. *Trichoderma virense* obtained from the soil [38] was received courtesy of Prof. Benjamin Horwitz (Technion—Israel Institute of Technology). A confrontation test (antagonism or mycoparasitism) was performed as described previously [8] by placing colony agar disks (6 mm in diameter) on a 90-mm diameter Petri dish containing a PDA. A 5-day *Trichoderma* spp. culture disk was placed in one pole of the plates in front of a similar disc taken from the 5-day culture margins of *M. maydis* (set in the opposite pole). The dishes were labeled and incubated at a temperature of 28 ± 1 °C in the dark. The hyphal growth was tested, and the interactions between the *M. maydis* pathogen and the *Trichoderma* isolates were recorded and photographed after three and ten days. *Trichoderma* isolates that restrict the pathogen’s growth or grew above the pathogen’s mycelium were marked as having mycoparasitic potential. Each *Trichoderma* isolate was tested in 5 independent repeats and similar results obtained. One representative plate was selected, photographed, and presented in the Results section.

### 2.3. Trichoderma *spp.* Secreted Metabolites Effect on Magnaporthiopsis Maydis Submerged Cultures

Five 6-mm mycelial disks were taken from the margins of 2-day colonies of selected *Trichoderma* isolates, seeded, and incubated in a rich liquid PDB substrate at a temperature of 28 ± 1 °C in the dark, at 150 rpm shaking, for 6 days. The growth medium of the liquid PDB *Trichoderma* spp. cultures were separated by filtration using a Buchner funnel and Whatman filter no. 3. The growth medium pH was measured and adjusted to 5.1 ± 0.2 (the pH of PDB medium) with NaOH. The liquid was filtered again using biofilter bottles (0.22-micron filter, BIOFIL 500-mL vacuum bottle filter, Indore, India) for sterilization. From the filtered liquid, 100 mL was poured into a sterile 250-mL Erlenmeyer bottle. To the liquid, a sterilized 6% glucose solution was added to a final concentration of 2%, identical to the amount of glucose in a standard PDB substrate. The control was PDB medium *M. maydis* cultures maintained at the same conditions. Five colony agar disks of *M. maydis* were added to each Erlenmeyer bottle, and the flasks were plugged with a breathable stopper and incubated at 150 rpm at 28 ± 1 °C in the dark. After 6 days, the fungus’ fresh and dry biomass were weighed, and the liquid substrate transparency (at a 450-nm wavelength) and pH were determined. Mycelial samples from each treatment were transferred under sterile conditions to solid rich substrate (PDA) plates and incubated for 6 days at 28 °C in the dark to test the fungus’ vitality. Each treatment was performed in 5 repetitions; the whole experiment was repeated twice, and similar results were obtained.

### 2.4. Trichoderma *spp.* Secreted Metabolites Effect on Magnaporthiopsis Maydis Solid Media Cultures

The effect of the *Trichoderma* spp. hydrolytic enzymes and metabolites on the growth of the maize pathogenic fungi was determined by a growth assay on the membranes, as previously described [41]. Petri dishes containing PDA were covered with Cellophane Membrane Backing #1650963 (Bio-Rad Laboratories, Hercules, CA, USA) and a 6-mm diameter *Trichoderma* spp. Agar culture disk was sown in the center of the plate. The plates were incubated under standard growing conditions (see Section 2.1) for two days, and at the end, the cellophane with the *Trichoderma* colony was removed. The plate was then sown with a 6-mm diameter *M. maydis* mycelial disk (prepared as described in Section 2.1). The colony’s growth rate was measured after six days compared to the fungus’s growth rate on the PDA substrate only. Each treatment was performed in 3 repetitions, and the whole experiment was repeated twice with similar results.

### 2.5. Biocontrol of Trichoderma *spp.* Secreted Metabolites in Infected Corn Sprouts

Thirty 2 L pots were filled with local peat soil from a field with a long history of LWD infection (Amir, Mehogi-1 plot, coordinates: 33°09’59” N 35°36’52” E) [4,12]. The soil was mixed with 30% Perlite No. 4 (to aerate the ground). The inoculation method also consisted of sterilized and *M. maydis*-infected wheat grains used to spread the disease pathogen in the soil one week before sowing (20 g of seeds per pot). Those seeds were previously incubated for 4 h in tap water, autoclave-sterilized, and infected with an *M. maydis* mycelial disk—5 disks per bottle containing 50 cm^3^ of sterilized seeds. The inoculated wheat seeds were incubated at 28 ± 1 °C for about 10 days (or until the fungus developed). In each pot, 5 corn seeds of the prelude cv. (sweet maize from SRS Snowy River seeds, Victoria, Australia, supplied by Green 2000 Ltd., Bitan Aharon, Israel) were seeded to a depth of 4 cm. The plants were grown in a growth room in an artificial light regime of 16 h and 8 h of darkness, with 45–50% humidity at 28 ± 3 °C. The pots were watered every two days with 100 mL of tap water per pot. 

The *Trichoderma* spp. liquid cultures’ secreted metabolites (filtrated growth fluid after pH adjustment; see Section 2.3.) were applied by irrigation (100 mL per pot), instead of the regular irrigation, and three intervals as follows: at the emergence above the soil surface (6 DAS), one week after the emergence (13 DAS), and two weeks after the emergence (20 DAS). The emergence percentages of all the treatments were evaluated after 6 days from sowing. After 47 days (V5 phenological stage, fifth leaf appearance), the plants’ root and shoot weights were measured. Each treatment was conducted in 5 repetitions (each repetition was a pot containing 5 sprouts). The experiment was performed twice, and similar results were measured.

### 2.6. Biocontrol of Trichoderma *spp.* Mycelia in Infected Corn Sprouts

Alternatively, to the application of *Trichoderma* metabolites against *M. maydis*, the biocontrol agent mycelium’s direct addition was done in potted sprouts. The experimental conditions and the *M. maydis* inoculation were done as in Section 2.5. The selected *Trichoderma* isolates were grown in a liquid PDB medium (see Section 2.3) for 6 days, filtered, chopped, and 1 g wet weight of mycelia added to each seed upon sowing. The emergence above the soil surface was determined at 6 DAS, and the plants’ root and shoot fresh weights were measured after 40 days. Additionally, at the experiment’s end (phenological stage V5, fifth leaf appearance), DNA was extracted from the root tissues, and a quantitative detection based on the qPCR of the fungal DNA was performed. Each treatment was conducted in 5 repetitions (each repetition of a pot containing 5 sprouts). The experiment was performed twice, and similar results were measured. The two most effective *Trichoderma* isolates, *T. asperelloides* (T203) and *T. longibrachiatum* (T7407), with inhibitory activity against *M. maydis*, were selected for the field condition trial conducted through an entire growth period.

### 2.7. Trichoderma *spp.* Effect in Pots, Over a Full Growing Season, under Field Conditions

This study examined the biocontrol potential of selected *Trichoderma* isolates against *M. maydis* over a whole growth period, in pots, in an open-air enclosure under field conditions. The disease symptoms, growth, and yield of the maize plants were tracked. The experiments aimed at a field conditions simulation, and the reason for using open-air pots (positioned in the field) with naturally infested soil, instead of sowing the plants directly in the field soil, was to allow enhancing the soil inoculum and to achieve, as much as possible, high and equable infection. The use of pots also allows better control of the water regime. It should be emphasized that pathogenicity experiments cannot lean upon natural soil infection alone, resulting in highly variable data. Even in a heavily infested area, the spreading of the pathogen is nonuniform [4,39]. The pathogen is dispersed in small quantities in the soil, and the disease spreading is not uniform in the field.

The experiment was performed at the Gadash experimental farm, located near the Gome junction, Upper Galilee, Hula Valley, Northern Israel (coordinates: 33°10’48.6” N 35°35’11.6” E), in the spring and summer of 2018 (24th May–14th August). The meteorological data recorded during the growing season were nearly optimal for the LWD burst: temperature 27.1 ± 5.2 °C, humidity 60.4 ± 18.0 %, soil temp of the top 5 cm 28.9 ± 7.4 °C, radiation 328.9 W/m^2^, precipitation 9.9 mm, and evaporation 724.2 mm (data—average ± standard deviation, according to the Israel Northern Research and Development meteorological station data, Hava-1).

Forty 10 L pots were filled with local peat soil from a field known to be infected with the LWD (Neot Mordechai, coordinates: 35°35’13” E 33°09’20” N) [11,15]. The soil was mixed with Perlite No. 4 (for the ground aeration) in a ratio of 2:1. Each treatment included ten independent replications (pots). All treatments were sown with Prelude cv. seeds (5 seeds per pot). Fertilization and insecticide treatments were applied according to the Israel Ministry of Agriculture Consultation Service (SAHAM) growth protocol. Watering was done by drip line (two droppers per pot) and controlled by a computerized irrigation system. The water irrigation regime was 5.4 L per pot/2 days for the first six days and 2.7 L per pot/2 days from 7 DAS onwards.

The plant inoculation and growth methodology were similar to that of Degani et al. 2019 and 2020 [4,8]. The inoculum method was the use of naturally infested peat soil, as described above, and a complementary inoculation with the Hm-2 isolate that was carried out in 2 steps. First, 40 g of sterilized infected wheat seeds were mixed with the top 20 cm of each pot’s soil with the seeding. These seeds were preincubated for three weeks at 28 °C in the dark with *M. maydis* 10 culture agar disks per 100 g seeds and used here to disperse the pathogen. Second, with the plants’ aboveground emergence (13 DAS), two *M. maydis* colony agar disks (6-mm diameter; see Section 2.1) were added to the upper parts of the roots (4 cm beneath the ground surface).

The experiments included 4 treatments: (1)Infected control—a naturally infested soil with the addition of complimentary *M. maydis* infection (as described above).(2)Similar infected soil to which *T. asperelloides* (isolate T.203) was added. The biocontrol agent was grown in a liquid PDB medium (see Section 2.3) for 6 days, filtered, chopped, and 1 g wet weight of mycelia added to each seed upon sowing.(3)Similar infected soil and *Trichoderma* treatments to (2) but with *T. longibrachiatum* (isolate T.7407).(4)Noninfected control—peat soil from the Gadash experimental farm (location detailed above), with no record of LWD but that carried similar characteristics to the infected field soil used in the other treatments. If such an infestation existed, it was assumed to be very low.

After 13 days, the germination percentages were calculated. At 34 DAS, the sprouts were thinned to one plant per pot, and the thinned plants were used to evaluate the plants’ growth indices. At the experiment end (82 DAS), the plants’ health status, height, fresh roots and shoots weight, and the yield (total cob weight) of the treatments were stated. The health assessment was based on four categories: healthy (4), minor symptoms (3), dehydrated (2), and dead (1). In addition, samples from the first aboveground internode of the plant stems were taken for DNA purification and qPCR.

### 2.8. Molecular Diagnosis

The qPCR diagnosis of *M. maydis* DNA in maize plants was made separately to the plants’ roots (in the growth room trials) or the first aboveground stalk’s internode (in the open-air full-growth experiment). The plants’ parts were washed with running tap water, then twice with sterile double-distilled water (DDW). The plant tissues were cut into a section of about 2 cm, and the weight of each repetition was adjusted to 0.7 g. The DNA isolation and purification were done according to the protocol of Murray and Thompson (1980) [42], with slight modifications as described earlier [39].

The DNA samples were stored at −20 °C and were used for the qPCR, as described previously [4]. This molecular method is based on a standard qPCR protocol used to detect mRNA (converted to cDNA) [43]. Instead, it has been optimized to detect the DNA of the pathogen *M. maydis* using species-specific primers [44,45]. The A200a primers were used for qPCR (sequences detailed in Table 2). The housekeeping gene, COX, encoding the enzyme cytochrome C oxidase (the last enzyme in the cellular respiratory electron transport chain in the mitochondria), aimed to normalize the *M. maydis* pathogen DNA [46]. This gene was amplified using the primer set COX F/R (Table 2). Calculating the relative gene abundance was according to the ΔCt model [47]. The same efficacy was assumed for all samples. All amplifications were performed in triplicate.

The real-time PCR reactions were executed using the ABI PRISM 7900 HT Sequence Detection System (Applied Biosystems, Waltham, MA, USA) and 384-well plates. Conditions of the qPCR were as follows: 5-µL total reaction volume was used per sample well—2 µL of DNA sample extract, 2.5 µL of iTaq™ Universal SYBR Green Supermix (Bio-Rad Laboratories Ltd., Hercules, CA, USA), 0.25 µL of forward primer, and 0.25 µL of reverse primer (to a well 10 µM from each primer). The qPCR cycle plan was as follows: pre-cycle activation phase, 1 min at 95 °C, 40 cycles of denaturation (15 s at 95 °C), annealing and extension (30 s at 60 °C), and finalizing by a melting curve analysis.

### 2.9. Statistical Analysis

A completely randomized statistical design was used when assessing the *Trichoderma* spp. and their secretion products outcome on the symptoms in the in vitro growth medium tests, in the growth chamber sprout infection, and in the full-growth season pot experiment under open-air conditions. Data analysis followed by statistics was done using the JMP program, 15th edition, SAS Institute Inc., Cary, NC, USA. The one-way analysis of variance (ANOVA) was used with a significance threshold of *p* < 0.05. The ANOVA analysis was followed by a post hoc of the Student’s *t*-test for each pair (without multiple comparisons correction). Ordinarily, in field condition experiments, molecular DNA measurements result in a high level of variations within the results due to variations in the host susceptibility, environmental conditions, and the spreading nature of the LWD pathogen [4,39]. Therefore, relatively high standard error values resulted in most of those tests, and statistically significant differences could hardly be identified.

## 3. Results

The use of pesticides to deal with the late wilt of maize in corn, which has been extensively researched in former studies, has the potential risk for environmental damage and health harm and is likely to induce pathogen resistance. In order to seek new ways of restricting *M. maydis*, we scanned *Trichoderma* spp. with bioprotective potential in an antagonism test that examined their ability to confront and suppress the maize pathogen (Figure 1). The confrontation assay results included the following possibilities: *M. maydis* mycoparasitism—the fungus was growing above the *Trichoderma* sp. colony surface. See, for example, T14707.*Trichoderma* mycoparasitism—the fungus was growing above the colony surface of the *M. maydis*. See, for example, T203.Antagonism—none of the two fungi can extend above the other, and their growth was stopped at the meeting point with the other fungus, usually producing a dark line. See, for example, T1607 and Tvir.

The results of this assay point at three *Trichoderma* isolates (Table 1 and Figure 1), *T. asperelloides* (T203), and *T. longibrachiatum* (T7407 and T.7507), which have a strong ability to grow on the *M. maydis* colony and prevent its spread. These three isolates, together with another more minor antagonist isolate, *Trichoderma* sp. T7107, were chosen for the next evaluation step—secreted metabolite influence on *M. maydis* growth in the submerged culture.

The *M. maydis* inhibition activity of those four selected isolates’ growth fluid was examined under controlled conditions for 6 days (Figure 2). In a preliminary experiment, the *Trichoderma* spp. growth in liquid PDB resulted in acidification of the medium (Figure 2A, gray bars). Interestingly, this change was especially evident in the three winners of the antagonism test present in Figure 1: *T. asperelloides* (T203, decrease of 1.8 pH units) and, in lesser severity, *T. longibrachiatum* (T7407 and T.7507, a reduction of 0.3 pH units). Thus, in the subsequent repeats (the results presented here), the *Trichoderma* spp. medium pH was adjusted to 5.1 (the pH of the PDB medium) before utilizing it in the assay. At the *Trichoderma* spp. metabolites biocontrol assay end, the pH values were measured again. After *M. maydis* growth, it was found almost unchanged in all the experimental groups.

The growth fluid metabolites of the isolate T.203, besides being the most acidic (before the pH correction), also resulted in the highest optical absorbance after being used as a medium for *M. maydis* submerged cultures (OD; Figure 2B). This high OD value resulted from the turbidity caused by a breakdown of the *M. maydis* hyphae into small fragments. In contrast, isolate T7107, a minor antagonist compared to the other isolates inspected here, had a negligible effect on the growth medium pH. It caused the lower OD level (i.e., it did not disrupt the *M. maydis* integrity). All *Trichoderma* isolates inspected at this stage excelled significantly (*p* < 0.05) in reducing the pathogen fresh and dry biomass after 6 days of growth (Figure 2C,D). Approximately 50% wet weight and about 80% dry weight decrease of the *M. maydis*, relative to the control, was recorded. None of these isolates stood out compared to the others in these two indices.

To test the inhibitory nature of the *Trichoderma* spp. secreted metabolites (whether they acted as a suppressor or fungicidal), one remaining *M. maydis* colony agar disk of each treatment was transferred to a fresh PDA plate. This enabled testing the vitality of the pathogen colonies. While the filtrates of the T7107, T.7407, and 7505T isolates were found to have the effect of fungicides, the pathogen sown in them did not recover; the T.203 isolate acted as an inhibitor, and after eliminating this factor and providing optimal conditions, the pathogen colony was able to regrow (Figure 3). This result was obtained even though T203 caused the fungal hyphae’s excessive discharge into many sections (Figure 2B).

The antagonistic capacity of the *Trichoderma* strains was also analyzed in a cellophane assay that allowed us to determine the effect of hydrolytic enzymes and metabolites secreted to the solid culture medium, as described before [41]. To this end, Petri dishes with PDAs covered with cellophane paper were used, and the isolates *T. asperelloides* (T203) and *T. longibrachiatum* (T7407 and T7507) were sown separately in the center of the plates. After allowing two days for the *Trichoderma* isolates to develop and their secretion products to accumulate in the medium, the cellophane (with the *Trichoderma* colony) was removed, and *M. maydis* was sown and incubated for an additional 6 days. Such a short preliminary *Trichoderma* growth is not enough to exhaust the essential nutrients in the substrate. It was found that the *Trichoderma* secretion products of all the isolates examined significantly (*p* < 0.05) reduced the pathogen’s development relative to the control. The best biocontrol result was obtained using isolate T7407, which reduced the growth rate 17-fold relative to the control (Figure 4).

The final stage in evaluating the *Trichoderma* spp. secreted metabolites was to add them to potted maize sprouts under controlled conditions. To this end, the *Trichoderma* isolates growth fluid was filtered and used, instead of regular irrigation, at three intervals (6, 13, and 20 DAS). The experiment was performed in natural field soil with a long history of late wilt infection, enriched with the *M. maydis* pathogen. As a preliminary step, we examined and found that the secretory products of *Trichoderma* isolates did not have an inhibitory effect on seed germination in the plates (Figure 5). Still, the impact of the substrate alone (PDB) on the rate of germination development was evident, apparently due to the high osmotic pressure. In potted, 47 days old plants, no significant effect was found in the *Trichoderma* isolates’ secretory products on the germination or the fresh biomass (weight of the roots and stalks; Table 3). However, irrigation with T7407 marked the highest value of the root weight (4.28 g, most close to the uninfected PDB control—5.98 g), while irrigating the *M. maydis*-infected sprouts with the PDB substrate marked the lowest weight of the root value (2.30 g).

Interestingly the shoot wet weight values fluctuated differently. While the shoot weight values were not statistically different from each other, they showed some opposite tendencies towards the roots (with the T7407 achieving the highest root weight and the lowest shoot weight results). The presence of the pathogen’s DNA in the roots at the experiment end was revealed using qPCR. It was found that irrigation with T203 and T7407 filtrates evidently reduced *M. maydis* DNA’s presence in the germ’s roots about five-fold and 1.9-fold, respectively, compared to the infected seeds in PDB medium (Table 3).

To summarize the experiment series with *Trichoderma* spp. liquid growth medium metabolites, it is evident that the T203 filtrate excelled as a bioprotective agent against the late wilt pathogen. For example, in the plate confront assay, the *M. maydis* submerged cultures assay (especially in degrading the pathogen hyphae) and in reducing the pathogen DNA in potted sprouts roots. Another promising candidate, the T7407 filtrate, excels in these assays in killing *M. maydis* in submerged cultures, inhibiting the pathogenic growth on solid PDA media and preventing a negative effect on potted sprout root wet weights. Therefore, those two *Trichoderma* isolates were chosen for the next experiment series: directly applied the biocontrol fungi in seedlings and a whole-growth period assay.

In corn sprouts (up to 40 days old, V5 phenological stage, fifth leaf appearance) grown on *M. maydis*-infected soil, the addition of the T203 isolate mycelial prevented the inhibitory effect of the pathogen on the sprouts’ emergence (Figure 6A). During this treatment, the emergence rate increased to 72% compared to the infected control that resulted in 40%. The T203 isolate biocontrol treatment significantly (*p* < 0.05, one-tailed *t*-test) improved the root biomass of plant sprouts grown in infected soil, which was 1.5 times higher than the infected untreated control (Figure 6B). Under the protection of this isolate (T203), the pathogen DNA in the plant roots decreased 23 times (Figure 6C) compared to the diseased plants in the control group. At the same time, isolate T7407’s protective influence achieved the highest result in the stalks’ fresh weight (Figure 6D), resulting in a significant (*p* < 0.05, one-tailed *t*-test) increase of nearly two times compared to the plants grown in infected soil without treatment.

The final experiment was designed to examine the two most successful *Trichoderma* isolates, *T. asperelloides* (T203) and *T. longibrachiatum* (T7407), mycelium effect in *M. maydis*-infected soil through an entire growing season. These two isolates passed all previous assays conducted in our isolate library, which gradually ruled out the ineffective ones. For this aim, the *Trichoderma* spp. were applied directly to potted plants with the sowing. Their biocontrol impact was evaluated under field conditions at 34 DAS (at the sprouting phase) and 82 DAS (the growth end).

Appraisal of the open-air potted plants (Table 4) revealed the bioprotective impact of isolate T203 on the appearance of the seedlings above the ground surface (84% emergence compared to the 64% of the untreated infected control, at 13 DAS). This isolate was less influencing at the sprouting phase (V5 phenological stage, 34 DAS) in regard to the sprouts’ growth parameters. Instead, isolate T7407 achieved 1.7-fold root fresh weight improvement compared to the infected control and a significantly (*p* < 0.05) higher shoot wet weight value than this control. The shoot height values’ fluctuations were minor, not significant at this stage, and were similar to all treatments.

At the growing season end (82 DAS), the *Trichoderma* isolates, *T. asperelloides* (T203) and *T. longibrachiatum* (T7407) achieved impressive results (Figure 7). The T203 isolate caused significant (*p* < 0.05) improvement in all the growth parameters relative to the infected control. These include the root and aboveground parts, fresh biomass, plants’ height, and yield. The average plants’ height significantly improved also in the T7407 isolate (Figure 7A,B). The qPCR analysis to evaluate the amount of pathogen DNA (Figure 7C) and the dehydration assessment (Figure 7D) supported the growth indices results. The *Trichoderma* isolates managed to reduce the amount of pathogen DNA in plant tissues by 40% (T203) and 96% (T7407) compared to the unprotected infected soil. The dehydration evaluation showed that T7407 mycelium enrichment of the seeds at sowing almost eliminated the disease symptoms, resulting in 60% healthy plants (compared to the non-infected control of 70% healthy plants). In comparison, the inoculated control registered only 20% healthy plants. The percentage of plants dried completely in both treatments T203 and T.7407 was 20%, half of the killed plants’ rate in the nonprotected inoculated treatment.

## 4. Discussion

Late wilt disease poses a serious risk to the corn industry in highly infected areas. Attempts have been made in the past to eradicate the *M. maydis* pathogen by chemical and biological means. The primary measure applied against the disease causal agent in Israel, Egypt, Spain, and India is resistant maize genotypes [16,49,50,51]. Genetic options are considered the most eco-friendly and economical for reducing grain yield losses [52]. As a short-term breeding strategy, even if moderately tolerant hybrids are identified and deployed for commercial production, it is possible to reduce the grain yield losses by about 0.2 tons per ha. This positive effect may increase to 3 tons per ha if tolerant hybrids are deployed [52]. Still, LWD resistance cultivars that are grown routinely may gradually lose their immunity to the disease due to the appearance of virulent pathogenic variations of *M. maydis* [31,32]. In recent years, dedicated research efforts to locate and implement a chemical pesticide against the LWD in Israel have resulted in encouraging results [4,15,28]. Commercial fungicide mixtures that contain Azoxystrobin can be applied through a dripline irrigation system at three 15-day intervals and achieve high protection against the LWD causal agent, even in a severe risk area and highly susceptible cultivars. This solution requires deploying drip-based irrigation lines to each row of plants, but the expenses can be reduced if the irrigation line is placed in the center of two adjacent rows [12].

However, intensive chemical treatment has several drawbacks in the short and long run. In the short run, the intense chemical treatment may lead to the emergence of fungicide resistance. Such cases have become increasingly common [53]. In the long-term, phytopathogenic fungi chemical restrains may cause human, animal, and environmental hazard risks. Reducing chemical fungicides has become more and more critical and is nowadays a global effort [29].

For these reasons, much research effort in the last two decades has been dedicated to seeking alternative ways to control LWD. The majority of those works focused on environmentally friendly substitutions to the traditional chemical fungicides. Briefly, these green approaches include: the effect of a tillage system and cover crop on maize mycorrhization and LWD [27]; siderophore production by *Bacillus subtilis* and *Pseudomonas koreensis* [34]; antagonistic other phytopathogens such as *Macrophomina phaseolina* [8]; agro-mechanical approaches such as excessive watering [54]; applying *Lycium europaeum* extracts [24]; manipulating maize plant’s growth hormones [55]; manipulating the root colonization by rhizobacteria; yeast and using organic compounds [56,57]; seed treatments with biocontrol formulations (*B. subtilis*, *B. pumilus*, *Pseudomonas fluorescens*, and *Epicoccum nigrum*); and bentocide, zinc oxide nanoparticles, and nano-silica [58]. As in other fungal pathogens, *Trichoderma* species have received particular focus in the research [22]. While some of these methods were tested under controlled conditions, many were applied in field trials and gained positive results. Stilt, the scientific effort towards revealing the potential of such control strategies, is continuing. Biological control has an important advantage since the biological agent changes along with the pathogen variations. A future integrative solution may combine such eco-friendly solutions with more traditional chemical control methods [29]. 

The current work results contribute to this global effort and suggest two *Trichoderma* species, *T. asperelloides* (T203) and *T. longibrachiatum* (T7407), as potential biocontrol interventions to reduce LWD severe economic losses. The secreted metabolites of these two species gained promising results in the *M. maydis* growth assays. However, their addition to pots in the form of a growth medium filtrate (after pH correction) was insufficient to significantly restrict the maize pathogen. Still, identifying antifungal active ingredients in the secreted crude metabolites, isolating them, and implementing them may result in a different and better outcome. Such a follow-up work may enrich our choice of options to fight the disease. It certainly may have a high value to farmers if the effort succeeds.

In contrast to the secretion metabolites trials, the direct application of the two *Trichoderma* species (T203 and T7407) as hyphae fragments to the seeds with sowing resulted in encouraging and significant results. These bioprotective agents prevent the pathogen from spreading in the host plant tissues, which was probably a major factor in rescuing the plants’ health, growth values, and yields (Figure 7).

In considering new applications based on the two promising *Trichoderma* species identified in the current work, it seems that combining T203 and T7407 in future treatments will be a good idea. Such a combination will bring about their unique beneficial impact and may result in synergistic protection against the late wilt pathogen. The different performances of those two isolates in the various experiments conducted here suggest that they have different action modes. Such differences are, for example, the ability of T203 to acidify the growth medium (PDB; Figure 1), to degrade the host fungous (*M. maydis*) hyphae into small fragments (OD assay; Figure 2A), and to suppress (but not kill) the host fungous in submerged cultures (Figure 3). In contrast, the T7407 isolate excels in the cellophane assay, and its secreted metabolites kill *M. maydis* in submerged cultures and significantly reduce its growth on a solid, rich medium. It was also able to restrain *M. maydis* DNA spread in maize plants in the field conditions over a full-growth period. Interestingly, T7407 and T7507, both strains of the same species, *T. longibrachiatum*, displayed different bioprotective behaviors against the late wilt pathogen in some of the tests. For example, they significantly differed in their ability to degrade the pathogen hyphae in the submerged cultures.

The results presented here encourage more in-depth research to uncover such biological agents’ potential and the methods to implement them in commercial fields. If adequately developed into final products and combined with other control methods, the *Trichoderma*-based control could play an essential role in maize crop protection against late wilt.

## 5. Conclusions

Late wilt disease is causing serious damage to maize fields throughout Israel. The disease is typified by the rapid wilt of sweet and fodder maize, mainly from the tasseling stage phase until shortly before maturity. The causal disease agent is the phytopathogenic fungus *M. maydis.* Over the past three years, we have scanned eight *Trichoderma* isolates as candidates for a biological control against *M. maydis*. Two of these isolates exhibited strong inhibitory activity against the maize pathogen: *T. asperelloides* (T203) and *T. longibrachiatum* (T7407, from a marine source). In vitro assays (dual plate culture), growth medium extract inhibition methods, and growth-room seedling pathogenicity trials all supported those isolates’ (or their secretions’) antagonistic effects against *M. maydis*. In this experiment series, the *Trichoderma* isolates positively improved the plants’ growth parameters, reduced the late wilt disease symptoms, and caused a prominent decrease in the host tissues’ pathogen’s DNA. Moreover, significant results were obtained during a full-season growth experiment (82 days) under field conditions. The current research results encourage further works that will focus on the isolation, characterization, and precise identification of the inhibitory components of T203 and T7407. These steps are essential for potential applications that can be applied in commercial fields against the *M. maydis* pathogen.

## Figures and Tables

**Figure 1 jof-07-00315-f001:**
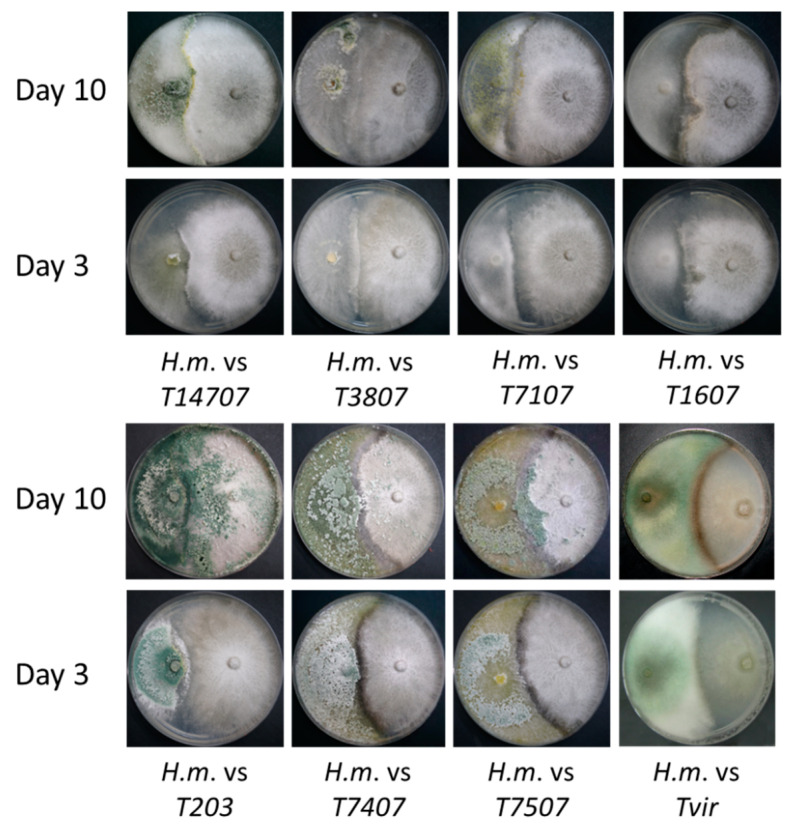
Plate mycoparasitism assays. The plate assay to identify the interactions between *Magnaporthiopsis maydis* and *Trichoderma* spp. in a rich potato dextrose agar (PDA) culture medium. The two fungi were planted opposite to each other, *Trichoderma* spp. on the left and *M. maydis* on the right. Images were taken at 3 and 10 days of growth.

**Figure 2 jof-07-00315-f002:**
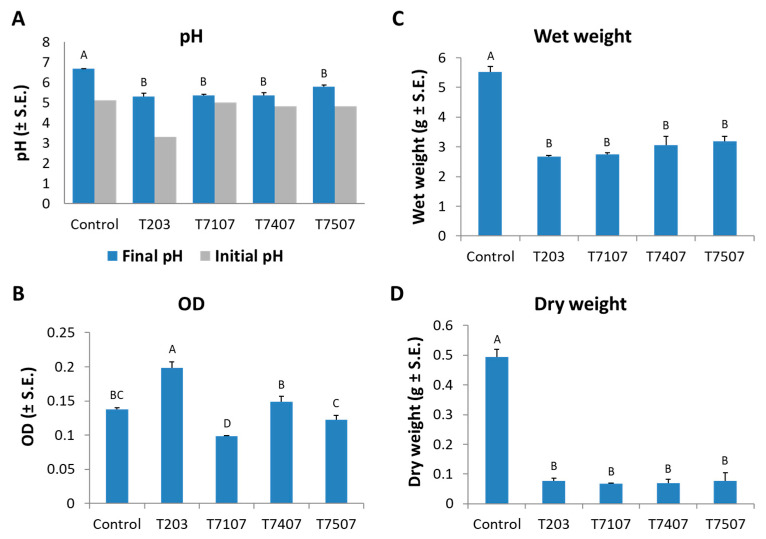
Effect of the growth fluid metabolites of *Trichoderma* isolates on the development of *M. maydis* submerged cultures. The *Trichoderma* species tested are listed in Table 1. All the colonies (*Trichoderma* spp. and *M. maydis*) were grown for 6 days. The control is potato dextrose broth (PDB) medium *M. maydis* cultures maintained at the same conditions. (**A**) Substrate acidity after *Trichoderma* spp. growth (gray bars) and after pH adjustment to 5.1 and the growth of *M. maydis* in those media (blue bars). At the experiment end (after *M. maydis* growth), the following parameters were measured: (**B**) The growth media turbidity (high optical density (OD) is the consequence of degradation of *M. maydis* mycelia into many fragments). (**C**) *M. maydis* wet weight. (**D**) *M. maydis* dry weight. The error lines represent a standard error of 5 repetitions (except for the initial pH bars in (**A**), the stock growth solution, i.e., one repeat). Statistical significance (*p* < 0.05) of variance was tested using a one-way ANOVA test and represented by different letters (A–D) above the chart bars.

**Figure 3 jof-07-00315-f003:**
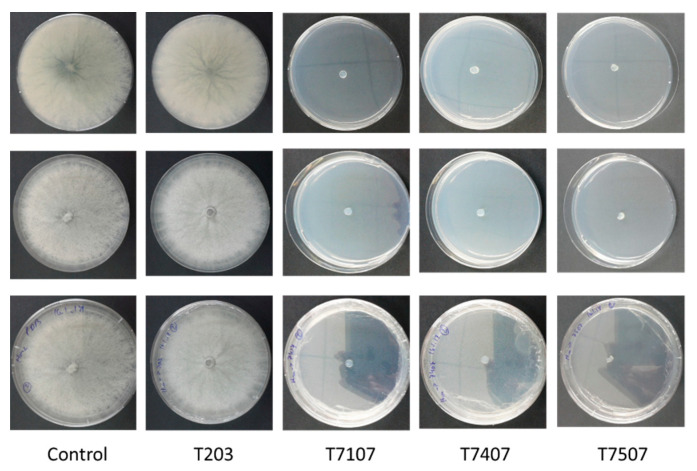
*M. maydis* vitality following incubation in the growth fluid of *Trichoderma* isolates. The *Trichoderma* isolates are *T. asperelloides* (T203), *Trichoderma* sp. (T7107), and *T. longibrachiatum* (T7407 and T7507). One disk of each of the treatments, described in Figure 2, was transferred to a PDA plate, and the plates were incubated for 6 days. The control is regular PDA medium *M. maydis* cultures maintained at the same conditions.

**Figure 4 jof-07-00315-f004:**
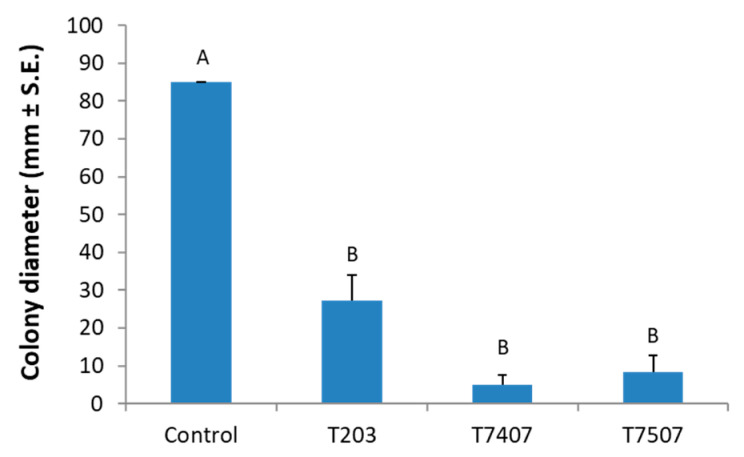
The *Trichoderma* spp. secreted metabolites effect on *M. maydis* solid media cultures. The *Trichoderma* isolates are *T. asperelloides* (T203) and *T. longibrachiatum* (T7407 and T7507). PDA dishes covered with cellophane membranes seeded with *Trichoderma* isolates for two days. After that, the cellophane with the *Trichoderma* colony was removed. The plate was then sown with *M. maydis* and incubated for 6 days. The control was regular PDA medium *M. maydis* cultures maintained at the same conditions. Values are the means from three biological replicates ± standard error. Different letters (A,B) indicate significant differences (one-way ANOVA test, *p* < 0.05).

**Figure 5 jof-07-00315-f005:**
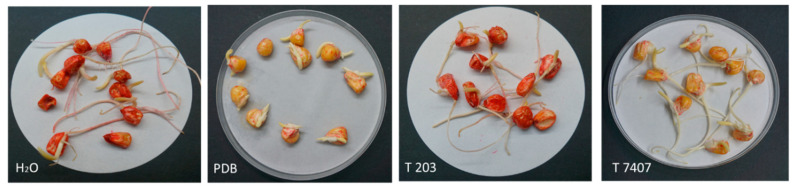
Effect of growth media of *Trichoderma* isolates on corn seed germination. The *Trichoderma* isolates are *T. asperelloides* (T203) and *T. longibrachiatum* (T7407). The seeds germinated in Petri dishes soaked in 4 mL of tap water (H_2_O), PDB, or PDB + secretion products (growth medium filtrate after 6 days *Trichoderma* isolates T203 or T7407 growth). Plate images are displayed after 5 days of incubation.

**Figure 6 jof-07-00315-f006:**
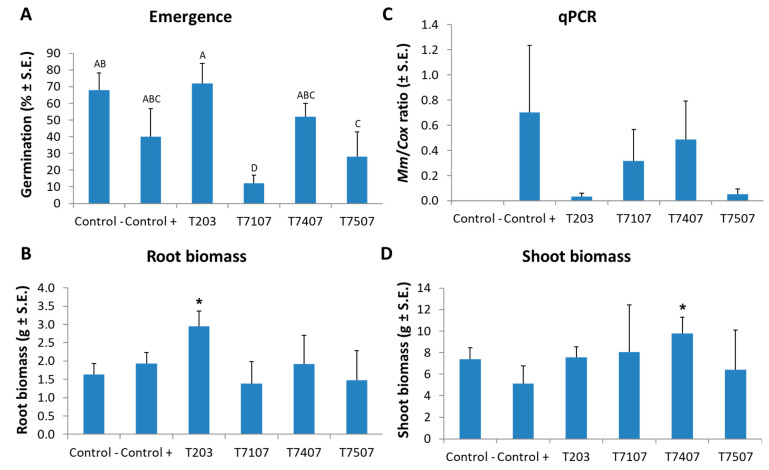
Maize sprout average growth indices at 40 DAS due to infection with *M. maydis* and *Trichoderma* spp. addition. The *Trichoderma* species tested are listed in Table 1. The emergence above the ground surface (**A**) was evaluated after 6 days. All other measures, roots’ wet biomass (**B**) and qPCR results (**C**), and stalk wet biomass (**D**), were determined at the experiment end. The real-time PCR (qPCR) results (**C**) are a relative amount of *M. maydis* DNA extracted from root samples normalized to the cytochrome C oxidase (COX) DNA. Control -, is soil without *M. maydis* infection. Control +, is the same soil with infection. All *Trichoderma* spp. treatments were conducted in such infected soil. The experiment was performed for 5 repetitions. Error lines represent a standard error. In (**A)**, a statistically significant (one-way ANOVA, *p* < 0.05) difference between the treatments was indicated by different letters (A–D). No statistical difference was identified using the one-way ANOVA test in (B, C, D). However, a one-tailed *t*-test compared to the infected control (which is a more powerful test) revealed significant differences (*p* < 0.05), indicated by an asterisk (*).

**Figure 7 jof-07-00315-f007:**
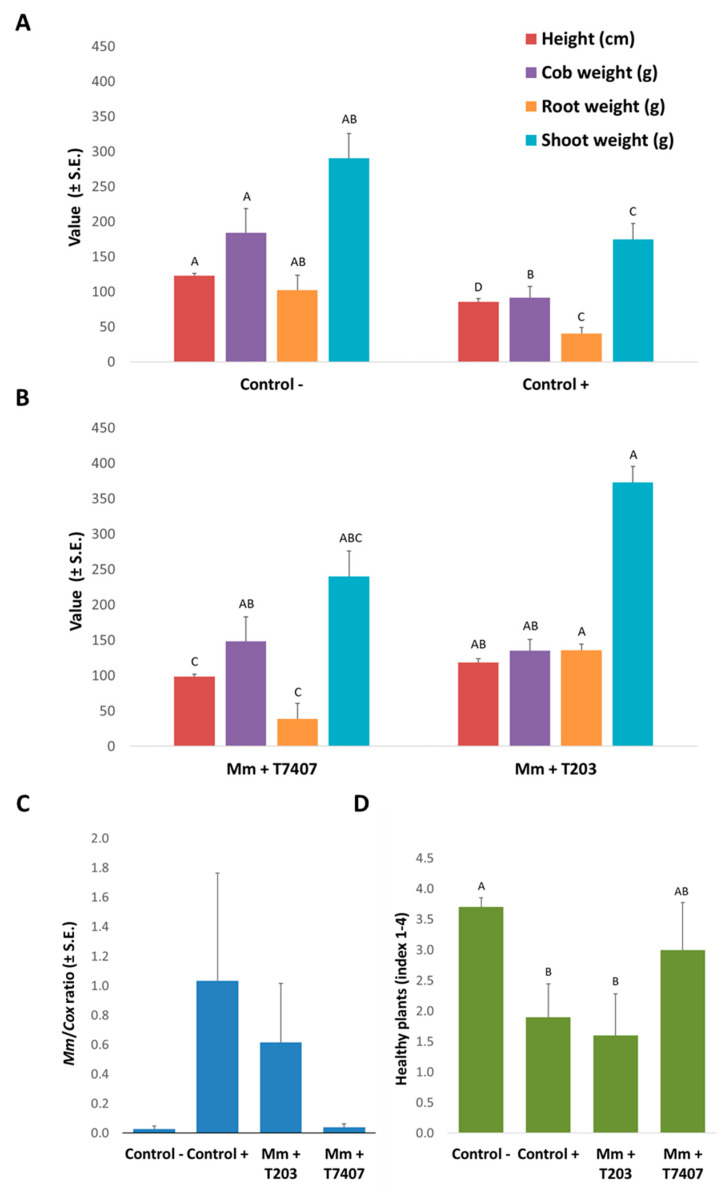
*Trichoderma* spp. effect on late wilt in pots over a full growing season under field conditions. Seeds of the susceptible maize genotype Prelude cv. were grown in *M. maydis*-infested soil to which *Trichoderma* isolates were added with sowing. Control −, is soil without *M. maydis* infection. Control +, is the same soil with infection. Growth parameters of the controls (**A**) and of the *T. asperelloides* (T203) and *T. longibrachiatum* (T7407) treatments (**B**) were measured at 82 DAS. qPCR diagnosis of *M. maydis* DNA (**C**) was made in the maize plants’ first aboveground internode. The y-axis values are the relative *M. maydis* DNA (*Mm*) abundance normalized to cytochrome C oxidase (*Cox*) DNA. The plants’ health state determination (**D**) is based on four categories: healthy (4), minor symptoms (3), dehydrated (2), and dead (1). The index presented is the mean calculated results. Error lines represent the standard error of the average of five replicates. Statistically significant (*p* < 0.05) differences between the treatments (at the same measures) are indicated by different letters (A–D).

**Table 1 jof-07-00315-t001:** Fungi used in this research and plate confrontation assay results.

Species	Designation	Origin	Reference	Confrontation Assay Winner ^2^	Tested in Sprouts	Tested in the Field
*Magnaporthiopsis maydis*	Hm2	*Zea mays*	[11,39]			
*Trichoderma Virense*	*Tvir* (IMI 304061)	Soil (Pantnagar, India)	[38]	Antagonism	No	No
*Trichoderma asperelloides*	T1607	*Psammocinia* sp. ^1^	[37]	Antagonism	No	No
*Trichoderma* sp. *O.Y. 7107*	T7107	*Psammocinia* sp. ^1^	[37]	Antagonism	No	No
*Trichoderma* sp. *O.Y. 14707*	T14707	*Psammocinia* sp. ^1^	[37]	*M. maydis*	No	No
*Trichoderma atroviride*	T3807	*Psammocinia* sp. ^1^	[37]	*M. maydis*	Yes	No
*Trichoderma longibrachiatum*	T7507	*Psammocinia* sp. ^1^	[37]	*T. longibrachiatum*	Yes	No
*Trichoderma longibrachiatum*	T7407	*Psammocinia* sp. ^1^	[37]	*T. longibrachiatum*	Yes	Yes
*Trichoderma asperelloides*	T203	ATCC 36042, CBS 396.92	[40]	*T. asperelloides*	Yes	Yes

^1^ Mediterranean sponge *Psammocinia* sp. ^2^ Confrontation assay results, including the following possibilities: *M. maydis* or *Trichoderma* sp. mycoparasitism (one of the fungi is growing above the colony surface of the other) and antagonism—none of the two fungi can extend above the other, and their growth was stopped in the meeting point with the other fungus, usually producing a dark line.

**Table 2 jof-07-00315-t002:** Primers for *Magnaporthiopsis maydis* detection.

Pairs	Primer	Sequence	Uses	Amplification	References
Pair 1	A200a-for A200a-rev	5′-CCGACGCCTAAAATACAGGA-3′ 5′-GGGCTTTTTAGGGCCTTTTT-3′	Target gene	200 bp *M. maydis* species-specific fragment	[11]
Pair 3	COX-F COX-R	5′-GTATGCCACGTCGCATTCCAGA-3′ 5′-CAACTACGGATATATAAGRRCCRR AACTG-3′	Control	Cytochrome C oxidase (*COX*) gene product	[46,48]

**Table 3 jof-07-00315-t003:** The effect of *Trichoderma* isolates secreted metabolites on the development of potted corn sprouts ^1^.

Growth Parameter	Control − DDW		Control − PDB		Control + PDB		T203 +		T7407 +		Control + DDW	
	Mean	SE.	Mean	SE.	Mean	SE.	Mean	SE.	Mean	SE.	Mean	SE.
Emergence (%) 10 DAS	52%	4.9%	60%	6.3%	56%	19.4%	60%	8.9%	76%	4.9%	68%	4.9%
Root wet weight (g)	3.79 ^B^	0.63	5.98 ^A^	1.05	2.69 ^B^	0.49	2.30 ^B^	0.42	3.75 ^B^	0.48	4.28 ^AB^	0.96
Shoot wet weight (g)	3.79	0.81	5.70	0.57	6.42	2.19	5.91	1.90	6.48	1.51	4.63	1.01
qPCR (*Mm*/*Cox* ratio)	0.01 ^B^	0.01	0.04 ^B^	0.04	1.37 ^A^	0.44	0.64 ^AB^	0.40	0.12 ^B^	0.05	0.34 ^B^	0.06

^1^ Maize sprouts average growth indices at 47 days after sowing (DAS) resulting from the infection with *M. maydis* and *Trichoderma* spp. Isolate secreted metabolite additions. The experiment was conducted in a growing room, and the *Trichoderma* isolates growth filtrate was used, instead of the regular irrigation, at three intervals (6, 13, and 20 DAS). Control −, is soil without *M. maydis* infection. Control +, is the same soil with infection. The real-time PCR (qPCR) results indicated *M. maydis* proportionate DNA normalized to the cytochrome C oxidase (COX) DNA. Values represent an average of 3–5 independent replications ± the standard error. Statistically significant (one-way ANOVA, *p* < 0.05) differences between the treatments, at the same measures, are indicated by different letters (A,B).

**Table 4 jof-07-00315-t004:** *Trichoderma* spp. influence in pots under field conditions, at the sprouting phase ^1^.

Growth Parameter	Control − ^2^		Control + ^3^		T203 +		T7407 +	
	Mean	SE.	Mean	SE.	Mean	SE.	Mean	SE.
Emergence (%) 13 DAS	74%	7.3%	64%	9.3%	84%	7.5%	68%	20.6%
Root wet weight (g)	4.1	0.99	3.0	0.52	2.7	0.42	5.1	0.86
Shoot wet weight (g)	23.8 ^AB^	5.40	18.4 ^B^	2.62	15.7 ^B^	3.59	34.9 ^A^	7.51
Shoot height (cm)	20.5	1.02	19.9	1.20	18.0	1.87	22.7	0.79

^1^ Maize sprouts growth indices at 34 DAS. The biocontrol agents were grown in a liquid PDB medium, filtered, chopped, and 1 g wet weight of mycelia added to each seed upon seeding. Values represent an average of ten replications ± standard error. When exist, statistically significant (*p* < 0.05) differences between the treatments (at the same measures) are indicated by different letters (A,B). ^2^ Noninfected (control −)—soil with minor levels of *M. maydis* infestation. ^3^ Infected (control +) —a naturally infested soil with the addition of a complimentary *M. maydis* infection.

## Data Availability

The datasets generated during and/or analyzed during the current study are available from the corresponding author upon reasonable request.

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
