# Peer review of "Trichoderma Biological Control to Protect Sensitive Maize Hybrids against Late Wilt Disease in the Field"

_jof, 2021, doi:10.3390/jof7040315_

Round 1

Reviewer 1 Report

Reviewer: 1

Ms. Ref. No.: jof-1183395

Authors: Ofir Degani *, Shlomit Dor

Specific notes:

TITLE

Nothing to comment

ABSTRACT

Line 11: delete ”-” before ”soil”

Line 27: Delete ”(P < 0.05)”

General comment, the abstract should be a summary of the entire manuscript, but it is currently very dense and long. I suggest the authors abbreviate and summarize

KEYWORDS

Lines 34 and 35: the keywords that appear are double or triple, in addition to appearing some of them in the title. I suggest the authors provide other keywords.

  1. Introduction

Line 88: “(data according to the Israel Organization of Crops and Vegetables)”, this source of information should be provided as one more reference within the manuscript.

  1. Materials and Methods

Line 117: M. maydis should appear in italics

Line 121: M. maydis should appear in italics

Lines 122 and 123: more details about the growth conditions of the fungi (Temperature, Humidity, ...)

Line 124: fungous??, please, correct.

Line 126: M. maydis should appear in italics

  1. Results

Line 365: M. maydis should appear in italics

  1. Discussion

Line 558: Bacillus subtilis and Bacillus pumilus should appears abreviatted.

  1. Conclusions

Line 606: Magnaporthiopsis maydis should appears abreviatted.

Lines 608 and 609: Trichoderma asperelloides and Trichoderma longibrachiatum should appears abreviatted.

FIGURES

Figure 2: I recommend that authors increase the letter size of the "x" and "y" axes.

I also recommend that the authors eliminate the value that appears above the error bar, it is a duplicate value, since it can be seen on the y-axis scale.

Figure 4: I also recommend that the authors eliminate the value that appears above the error bar, it is a duplicate value, since it can be seen on the y-axis scale.

Figure 5: I recommend that authors increase the letter size of the "x" and "y" axes.

I also recommend that the authors eliminate the value that appears above the error bar, it is a duplicate value, since it can be seen on the y-axis scale.

Figure 6: I recommend that authors increase the letter size of the "x" and "y" axes.

I also recommend that the authors eliminate the value that appears above the error bar, it is a duplicate value, since it can be seen on the y-axis scale.

FIGURE LEGEND

Line 339 (Figure 1): “at 28°C ± 1 in the dark”, put in material and methods.

Lines 390-392 (Figure 2): “The dishes were maintained for 6 days at 28°C ± 1 in the dark to test the fungus’ vitality. Three repetitions for each Trichoderma isolate are displayed. Control is PDA medium M. maydis cultures, maintained at the same conditions”, put in material and methods.

Line 437 (Figure 5): “at 28°C ± 1 in the dark”, put in material and methods.

TABLES

Nothing to comment

REFERENCES

Nothing to comment

Author Response

Responses to the reviewer 1 comments

We thank the reviewer for investing substantial efforts, which are undoubtedly contributing to this manuscript. The remarks and suggestions improved this paper’s scientific soundness and accurateness. Your contribution is greatly appreciated.

  1. ABSTRACT

Line 11: delete ”-” before ”soil”

Deleted as per the reviewer’s advice.

Line 27: Delete ”(P < 0.05)”

Deleted as suggested by the reviewer.

General comment, the abstract should be a summary of the entire manuscript, but it is currently very dense and long. I suggest the authors abbreviate and summarize.

The reviewer is correct. The abstract was edited, shortened and we believe it is now more focused and more clear. 

  1. KEYWORDS

Lines 34 and 35: the keywords that appear are double or triple, in addition to appearing some of them in the title. I suggest the authors provide other keywords.

We agree. All the keywords that already appear in the title and some of the other keywords were deleted or replaced.

  1. Introduction

Line 88: “(data according to the Israel Organization of Crops and Vegetables)”, this source of information should be provided as one more reference within the manuscript.

A new reference was added to the text, as suggested by the reviewer (lines 85-87): “The extent of corn crops in metric ton yield per metric hectare in this country is exhibiting a constant upward trend, from 17.5 in 1987-1996 to 18.0 in 1997-2006 and 20.1 in 2007-2016 [33].”

  1. Materials and Methods

Line 117: M. maydis should appear in italics

Corrected to italics as advised.

Line 121: M. maydis should appear in italics

Corrected to italics as advised.

Lines 122 and 123: more details about the growth conditions of the fungi (Temperature, Humidity, ...)

We agree and therefore made the required changes to the text that now reads: “The isolate was grown on potato dextrose agar (PDA, Difco Laboratories Detroit, Michigan, USA) at a temperature of 28 ± 1 °C in the dark. These growth conditions allow a high humidity atmosphere inside the Petri dishes.” (lines 120-123).

Line 124: fungous??, please, correct.

This is a typo. It was corrected to fungus.

Line 126: M. maydis should appear in italics

Corrected as advised.

  1. Results

Line 365: M. maydis should appear in italics

Corrected as advised.

  1. Discussion

Line 558: Bacillus subtilis and Bacillus pumilus should appears abbreviated.

Corrected to B. subtilis and B. pumilus as advised.

  1. Conclusions

Line 606: Magnaporthiopsis maydis should appears abbreviated.

Corrected to M. maydis as advised.

Lines 608 and 609: Trichoderma asperelloides and Trichoderma longibrachiatum should appears abbreviated.

Corrected to T. asperelloides and T. longibrachiatum as advised.

  1. FIGURES

Figure 2: I recommend that authors increase the letter size of the “x” and “y” axes.

I also recommend that the authors eliminate the value that appears above the error bar, it is a duplicate value, since it can be seen on the y-axis scale.

We agree, and the following changes were subsequently made. The letter size of the “x” and “y” axes increased from 12 pt. to 14 pt. The axes headline font was increased from 14 pt. to 16 pt. We also removed all the values that appear above the error bars, as suggested.

Figure 4: I also recommend that the authors eliminate the value that appears above the error bar, it is a duplicate value, since it can be seen on the y-axis scale.

 The values that appear above the error bars were removed as suggested.

Figure 5: I recommend that authors increase the letter size of the “x” and “y” axes. I also recommend that the authors eliminate the value that appears above the error bar, it is a duplicate value, since it can be seen on the y-axis scale.

Figures numbers were updated. This is now Figure 6. The letter size of the “x” and “y” axes increased from 12 pt. to 14 pt. The axes headline font was increased from 14 pt. to 16 pt. We also removed all the values that appear above the error bars, as suggested.

Figure 6: I recommend that authors increase the letter size of the “x” and “y” axes. I also recommend that the authors eliminate the value that appears above the error bar, it is a duplicate value, since it can be seen on the y-axis scale.

 Figures numbers were updated. This is now Figure 7. The letter size of the “x” and “y” axes increased from 14 pt. to 16 pt. The axes headline font was increased from 16 pt. to 18 pt. We also removed all the values that appear above the error bars, as suggested.

  1. FIGURE LEGEND

Line 339 (Figure 1): “at 28°C ± 1 in the dark”, put in material and methods.

This information was already detailed in the Materials and methods. It was deleted from the Figure 1 legend as suggested.

The same correction was made to the Figure 2 legend.

Lines 390-392 (Figure 3): “The dishes were maintained for 6 days at 28°C ± 1 in the dark to test the fungus’ vitality. Three repetitions for each Trichoderma isolate are displayed. Control is PDA medium M. maydis cultures, maintained at the same conditions”, put in material and methods.

This information was already detailed in the Materials and methods. It was deleted from Figure 3 legend as suggested.

Line 437 (Figure 5): “at 28°C ± 1 in the dark”, put in material and methods.

This information was already detailed in the Materials and methods. It was deleted from Figure 5 legend as suggested.

Reviewer 2 Report

Row 117 should be italic M. maydis

Row 126 should be italic M. maydis

Row 232 not uniform or uniform? Even in a heavily infested area, the  spreading of the pathogen is uniform

Congratulations to authors!

Author Response

Responses to the reviewer 2 comments

We would like to express our sincere appreciation to the reviewer for essential and helpful advice. The time and effort invested are greatly appreciated and certainly contributed to the manuscript and improved it. Thank you.

  1. Row 117 should be italic maydis

Corrected to italics as advised.

  1. Row 126 should be italic maydis

Corrected to italics as advised.

  1. Row 232 not uniform or uniform? Even in a heavily infested area, the spreading of the pathogen is uniform.

This is a typo. It was corrected to nonuniform.
